# Comparison of Swim-Up and Microfluidic Sperm Sorting Methods in Selection of Sperm for Intracytoplasmic Sperm Injection

**DOI:** 10.3390/ijms26115374

**Published:** 2025-06-04

**Authors:** Michal Ješeta, Adéla Doubravská, Jana Antalíková, Lenka Mekiňová, Kateřina Franzová, Kateřina Remundová, Jan Hošek, Bartosz Kempisty, Robert Hudeček

**Affiliations:** 1Department of Gynecology and Obstetrics, Faculty of Medicine, University Hospital Brno, Masaryk University Brno, Jihlavská 20, 625 00 Brno, Czech Republic; doubravska.adela@fnbrno.cz (A.D.); mekinova.lenka@fnbrno.cz (L.M.); franzova.katerina@fnbrno.cz (K.F.); remundova.katerina@fnbrno.cz (K.R.); hudecek.robert@fnbrno.cz (R.H.); 2Laboratory of Reproductive Physiology, Institute of Animal Biochemistry and Genetics, Centre of Biosciences, Slovak Academy of Sciences, Dúbravská Cesta 9, 840 05 Bratislava, Slovakia; jana.antalikova@savba.sk; 3Veterinary Research Institute, Hudcova 70, 621 00 Brno, Czech Republic; jan.hosek@vri.cz; 4Department of Veterinary Medicine, Nicolaus Copernicus University, Gagarina 7 Str., 87-100 Toruń, Poland; 5Department of Human Morphology and Embryology, Wroclaw Medical University, Chałubińskiego 6a Str., 50-368 Wroclaw, Poland; 6College of Agriculture and Life Sciences, Nort Carolina State University, Raleigh, NC 27695, USA

**Keywords:** infertility, spermatozoa, IVF/ICSI, sperm analysis, sperm separation

## Abstract

The use of microfluidic sperm sorting (MFSS) systems in infertility treatment is increasing due to their practicality and ease of use. While often presented as highly effective, their efficacy in patients with varying sperm analysis results remains uncertain. In this study, we evaluated the effectiveness of MFSS compared with the swim-up (SU) technique in terms of oxygen radical levels and spermiogram parameters. Samples from each patient were processed using both methods, followed by assessments of sperm concentration, motility, morphology, DNA integrity, acrosomal status, and mitochondrial membrane potential. Participants were selected based on sperm analysis and categorized as normozoospermic (n = 40) or non-normozoospermic (n = 28). An analysis of separation techniques revealed no significant differences, except for a lower percentage of DNA-fragmented sperm in the MFSS group compared with SU within the non-normozoospermic cohort (SU: 10.0% vs. MFSS: 5.69%, *p* = 0.027). No differences were observed between SU and MFSS in normozoospermic men. The MFSS method is a simple technique, frequently used in laboratories, that yields good results but does not offer a substantial advantage over SU. The primary benefit of MFSS appears to be a significant reduction in the proportion of sperm with DNA fragmentation compared with SU in patients with abnormal sperm analysis results.

## 1. Introduction

The efficiency of human reproductive treatments over the last 10 years has not increased dramatically, despite significant advancements in methods and the development of various add-on approaches. The chance of achieving a healthy baby is only approximately 5% per retrieved oocyte [1]. One potential reason for the relatively low efficiency of in vitro fertilization (IVF) is the inability to select sperms with sufficient precision. During in vivo fertilization, sperm selection is highly rigorous at multiple levels of the female reproductive system, guided by natural navigation systems, such as thermotaxis, chemotaxis, or rheotaxis [2]. Of several hundred million at the beginning, only a few hundred sperms reach the immediate vicinity of the oocyte [3], and only one sperm reaches the cytoplasm of the oocyte. This means that there is usually a huge surplus of sperm and there is a lot of room for selection. Sperm separation in vivo is, therefore, very intense and is a multistep process that starts immediately after ejaculation at the level of the cervix uteri [4] and continues to the fine selection processes in the oviduct [5]. Spermatozoa populations exhibit significant heterogeneity, with only a small subset within the ejaculate retaining their capacity for fertilization [6].

Multiple approaches have been proposed for in vitro sperm separation. Unfortunately, the current methods for sperm selection are not perfect and do not allow selection for the specific subpopulation that is best for IVF. This could be attributed to several factors. Frequently, this aspect of IVF is underestimated in clinical settings, and because of the efficiency of the process and the ease of specimen handling, the most straightforward method is often selected. Although this approach yields satisfactory results and is often used, it is predominantly based on the principle of sperm selection according to motility. During in vitro selection, we attempted to make the best selection of sperms for fertilization. As the intracytoplasmic sperm injection (ICSI) method is increasingly being used, the importance of sperm selection is high, and poor sperm selection can have a fatal impact on fertilization success and embryonic development [7,8]. The most commonly used techniques for sperm selection for ICSI are swim-up (SU) and density gradient centrifugation (DGC) [7]. These methods isolate motile sperm with normal morphology [7,9]. However, these methods are not perfect and centrifugation may have a negative impact on sperm viability and can lead to sperm DNA fragmentation [10,11]. Indeed, conventional SU without centrifugation is known to be effective in reducing DFI [11]. Thus, the SU protocol involving short centrifugation is most commonly used in andrology laboratories that use insufficient yields when processing heavy oligozoospermic samples [9,12].

The inefficient selection of defective sperm or sperm with DNA fragmentation can harm fertilization, embryonic genome activation, and overall embryonic development [13,14]. Traditional separation techniques, such as SU or DGC, are gradually being replaced by newer microfluidic sperm sorting (MFSS) systems. These systems offer advantages such as simplicity, avoidance of centrifugation, ease of use, and improved selection outcomes when applied in human IVF [15,16,17]. According to one study, the use of the MFSS method significantly reduced the proportion of sperm with fragmented DNA from >30% to almost undetectable levels [16]. MFSS-based techniques are often considered more efficient for sperm selection sorting in ART than conventional SU or DGC methods [15,18,19]. After MFSS separation, a significantly higher embryo euploidy rate was detected in non-normozoospermic patients, in contrast to the swim-up method [15]. When comparing MFSS and density gradient sorting, significantly higher motility, normal morphology, and lower DNA fragmentation were found in normozoospermic and non-normozoospermic patients [20].

However, these methods may not be universally applicable for all infertility treatments. For instance, when comparing MFSS and SU for intrauterine insemination (IUI), some studies have reported superior outcomes with the SU method [21]. MFSS techniques are now increasingly used for sperm sorting. Sometimes, they present amazing efficiency compared with standard separation methods such as SU. In the past, the high benefit of MFSS methods over SU for separation has been manifest mainly in terms of concentration and motility, also leading to better fertilization and clinical pregnancy rates. However, there are studies that are not so optimistic and consider the importance of MFSS separations as insignificant and without effective impact on treatment success [22].

The aim of the current study was to conduct a comprehensive comparison of a microfluidic sperm sorting system and the swim-up method. Throughout the comparison, we carefully monitored the oxygen radical content in seminal plasma to prevent secondary damage to sperm during separation. Additionally, we evaluated the effectiveness of both separation techniques in patients with varying sperm analysis results to determine their relevance in normozoospermic and non-normozoospermic individuals.

## 2. Results

### 2.1. Patients

A total of 68 patients from the CERMED center (part of the Department of Gynecology and Obstetrics, University Hospital Brno, Czech Republic) with a mean age of 32.14 years were included in this study. The average values of the evaluated parameters are presented in Table 1.

### 2.2. Results of Semen Analyses

After both separation methods, a significant decrease in sperm concentration was observed, along with an increase in motility and a higher proportion of morphologically normal spermatozoa (Table 2 and Figure 1).

#### 2.2.1. Sperm Concentration

Sperm concentration is an important parameter; however, it is not the sole determinant of efficiency. The key factor is the effectiveness of the separation system, even in patients with a low initial sperm concentration in the native ejaculate. The mean sperm concentration before separation was 44.57 million/mL. After separation, the sperm concentration significantly decreased, to 7.29 million/mL for the SU method and to 8.55 million/mL for the MFSS method (Table 2 and Figure 1).

#### 2.2.2. Total Sperm Count

The total sperm count in the ejaculate and the total number of spermatozoa obtained after separation are critical parameters. Before separation, the total sperm count was 165.58 million. During the separation process, this count significantly decreased, reaching 3.39 million/mL for the SU method; (1:48 ratio) and 3.98 million/mL for the MFSS method (1:42 ratio) (Table 2 and Figure 1). However, despite this reduction, these values remained sufficient for a successful IVF cycle. However, in patients with low total sperm count, such as oligozoospermia, this significant reduction may be a problem. It was reported that in ejaculates with a concentration below 10 million, one MFSS technique (Fertile Plus) did not work well [18].

Because it is not possible to use the entire ejaculate, but only a part of it, it may ultimately happen that not enough sperm are obtained for the final selection before ICSI. For ICSI, a minimum number of 100,000 sperms after separation is generally considered a threshold for optimal fertilization. While ICSI can be successful with lower sperm counts, fertilization and pregnancy rates may be reduced compared with higher sperm counts. While ICSI allows fertilization with a single sperm, the selection process aims to improve fertilization and embryo quality. We compared the success of separation in patients with low concentrations (<10 mil/mL) and the effectiveness of the swim-up technique; only one SU patient had less than 0.1 mil/mL, while in the MFFS group, there were four patients (Appendix A).

#### 2.2.3. Sperm Motility

Total motility increased significantly from 53.05% before separation to 91.29% after the SU method and 93.57% after the MFSS method. No statistically significant differences were observed between the two separation techniques (Table 2 and Figure 1).

Comparing the efficiency of separations in patients with very low concentrations (<10 mil/mL), it was found that after separation by the MFSS technique there were four patients who did not have 0.1 mil/mL sperm after separation and of these, three patients also had very low motility (they were classified as oligoasthenoteratozoospermia). With the swim-up technique, there was only one patient with <0.1 mil sperm, also with low motility (oligoasthenoteratozoospermia).

#### 2.2.4. Sperm Morphology

Due to the rigorous selection process during separation, sperm morphology improved significantly. Before separation, the proportion of sperm with normal morphology was 5.3%. After separation, this proportion increased significantly to 15.1% with the SU method and 15.6% with the MFSS method. No statistically significant differences were found between the two methods regarding the proportion of morphologically normal sperm (Table 2 and Figure 1).

#### 2.2.5. DNA Integrity

DNA integrity, which is not typically assessed in a standard spermiogram, plays a crucial role in oocyte fertilization and subsequent embryonic development. Currently, DNA integrity abnormalities are considered as one of the most common causes of idiopathic infertility. Following sperm separation using the SU method, the DNA fragmentation index (DFI) was significantly reduced by an average of 47.5%, resulting in a final DFI value of 9.57. After processing with the MFSS, the DFI value decreased further to 6.28.

Thus, MFSS led to a statistically significant reduction in DFI, not only compared with the unprocessed ejaculate, but also compared with the SU method.

#### 2.2.6. Acrosomal Status

The effect of sperm sorting on acrosomal status is presented in Table 2. Before separation, the percentage of sperm with an intact acrosome was relatively low at 29.15%. After separation, this percentage significantly increased to 83.32% (with SU) and 84.01% (with MFSS). However, no statistically significant difference was observed between the two separation methods (Figure 2).

#### 2.2.7. Mitochondrial Activity

The mitochondrial membrane potential (MMP) status following sperm sorting is presented in Table 2. Before separation, the percentage of spermatozoa with intact MMP was 87.62%. After sperm sorting, this percentage significantly increased to 93.2% with the swim-up (SU) method and 93.23% with the microfluidic sperm sorting system (MFSS), compared with the sperm before separation. However, no statistically significant differences were observed between spermatozoa selected by SU and MFSS (Table 2).

### 2.3. Detection of ROS/RNS in Seminal Plasma

It has previously been reported that hydrogen peroxide (H_2_O_2_), a key member of the reactive oxygen species family, negatively impacts sperm function by reducing viability, motility, the ability to penetrate cervical mucus, membrane integrity, and capacitation. However, these detrimental effects occurred only at concentrations exceeding 200 µM. Concentrations below this threshold do not significantly affect sperm function [23]. We measured the total reactive oxygen species (ROS) and reactive nitrogen species (RNS) in our cohort, including H_2_O_2_ as the primary ROS component. The average detected level was 47.25 µM (expressed as H_2_O_2_ equivalents, which served as a standard for total ROS/RNS determination), with a maximum value of 133.3 µM (H_2_O_2_ equivalents) (Appendix A). These values were well below the reported 200 µM threshold for negative sperm effects. It is important to note that samples were collected and frozen 60 min after ejaculation, and during this time, ROS/RNS concentrations declined naturally. Specifically, H_2_O_2_ levels decreased to 73.5% of their original values after 60 min of liquefaction (Appendix A). Thus, even considering this reduction, the predicted ROS/RNS level for the sample with the highest measured value (133.3 µM H_2_O_2_ equivalents) was approximately 181.3 µM after ejaculation, which was still below the 200 µM threshold.

### 2.4. Normozoospermic and Non-Normozoospermic Patients

Based on our previous experience and recent articles [22], we compared the effectiveness of sperm separation techniques in normozoospermic patients (n = 40) and non-normozoospermic patients (n = 28). The latter group included six asthenozoospermic, six oligoasthenoteratozoospermic, six teratozoospermic, four oligoteratozoospermic, two oligoasthenozoospermic, two oligozoospermic, and two asthenoteratozoospermic patients. In normozoospermic patients, no statistically significant differences were observed between the separation techniques across the analyzed parameters (Table 3).

Similarly, in the non-normozoospermic group, most parameters showed no significant differences between methods. However, a statistically significant difference was found in the proportion of spermatozoa with fragmented DNA. After MFSS separation, the proportion of sperm with fragmented DNA was significantly lower compared with the SU method (Table 4).

## 3. Discussion

In this study, we compared the effectiveness of the swim-up (SU) and micro-fluidic sperm sorting (MFSS) methods in normozoospermic and non-normozoospermic patients. To exclude the potential influence of reactive oxygen species (ROS) on sperm quality, we monitored oxygen radical concentrations in seminal plasma before patient inclusion. High ROS levels in seminal plasma are known to damage sperm and increase the proportion of sperm with fragmented DNA [24,25]. However, in our study, only low ROS levels were detected, which should not have significantly affected the evaluated semen parameters. This study was limited to in vitro analyses only. As this was a laboratory study using discarded, unidentified post-spermiogram samples, clinical data on fertilization success and subsequent embryonic development were not available. To verify our hypothesis, the patients’ identities were not crucial; rather, we focused on testing the effectiveness of the separation system across patients ranging from normal to infertile.

We found that the sperm concentration decreased significantly after each of the separation methods, although no differences were observed between MFSS and SU. Similarly, the total sperm count did not differ between methods. As expected, both techniques significantly increased the proportion of motile sperm compared with that of raw ejaculate. However, no statistically significant difference was found in the total motility between MFSS and SU. The variability in motility differences following separation depended largely on how the SU method was performed, as some studies reported significant differences between MFSS and SU [26], whereas others, including ours, showed minimal differences [15]. Both methods also led to a significant improvement in sperm morphology compared with that of raw semen. However, we found no significant difference between MFSS and SU in terms of the proportion of morphologically normal sperm. These results align with those of previous studies [15], although they differ from findings where density gradient centrifugation (DGC) was compared with MFSS, revealing a significantly higher proportion of normal spermatozoa in the MFSS group [19].

A key finding of our study was that MFSS significantly improved the proportion of sperm with non-fragmented DNA, compared with SU. It is likely that the strict selection process of MFSS contributed to this outcome, as there appears to be an inverse relationship between DFI and motility. MFSS separation enhances the selection of spermatozoa without DNA fragmentation, which could ultimately lead to healthier embryo development [27]. Previous reports have also demonstrated that MFSS strongly reduces the proportion of sperm with fragmented DNA, compared with both DGC [16,28] and SU [29].

In our study, MFSS significantly improved the proportion of spermatozoa with non-fragmented DNA, but only in non-normozoospermic patients. No significant differences were observed between MFSS and SU in normozoospermic patients. This finding is consistent with previous research showing that MFSS separation significantly reduces embryo aneuploidy in non-normozoospermic patients but not in normozoospermic patients [2]. Additionally, studies comparing MFSS and SU separation have found similar results regarding fertilization potential and DNA fragmentation in both normozoospermic and non-normozoospermic patients [17]. While gentle centrifugation has a minimal impact on normal sperm samples, semen samples with abnormalities appear to be more susceptible to mechanical stress and DNA damage [30,31]. This observation could explain why MFSS, a centrifugation-free technique, reduces DNA fragmentation rates, particularly when processing semen samples with suboptimal parameters.

For successful fertilization, sperm must be motile with active mitochondria, have intact acrosomes, and be capable of undergoing an acrosomal reaction before penetrating the egg [32]. For these reasons, we decided to detect sperm condition, morphology, mitochondrial status, and acrosome status as parameters of sperm condition after separation, so that we could detect any possible sperm damage during separation. In this study, we found no significant difference in acrosomal status between the two separation methods. However, both techniques significantly increased the proportion of sperm with intact acrosomes compared with pre-separation conditions. Notably, the values obtained were higher than those reported in older studies [33].

We also assessed mitochondrial membrane potential (MMP) using the fluorescent marker JC-1. As mitochondria generate ATP in the midpiece of the sperm flagellum, they play a crucial role in sperm energy production and motility. Mitochondrial status is a maintaining factor for chromatin integrity, motility, and acrosome reaction [31]. Our findings showed that MMP was lower in raw ejaculate than in both separation techniques, and there was no significant difference between MFSS and SU. Sperm mitochondrial activity correlates well with progressive motility and acrosomal status, and our results are consistent with those of studies that reported a positive correlation between MMP and motility [31,34].

To exclude the impact of oxidative stress, we measured oxygen radical levels in the seminal plasma before separation. High ROS levels can damage sperm membranes, impair motility [35], and contribute to high DNA fragmentation rates [24,25]. In our cohort, the detected ROS values ranged from 8 to 133 µM, all of which were below the 200 µM threshold associated with sperm damage [23]. Furthermore, correlation analysis showed no direct relationship between ROS levels and the observed semen parameters (Figure 3C), confirming that ROS did not significantly affect the results of our study.

Previous studies have reported mixed results when comparing MFSS to standard sperm sorting methods. Some studies found notable benefits [16,26,28], whereas others reported only marginal improvements with no statistically significant differences, as highlighted in a recent meta-analysis [36]. In the present study, MFSS significantly reduced DNA fragmentation in non-normozoospermic patients. A key limitation of MFSS is that it yields fewer sperm after separation due to the maximum ejaculate volume limit of 1.0 mL. While this is not a concern in normozoospermic patients, it can be problematic for individuals with low semen volume or sperm concentration. Therefore, the choice of sperm separation method should be individualized based on the patient’s characteristics.

Although microfluidic sperm sorting is considered the future of assisted reproduction, its development has remained stagnant for two decades [37]. However, in recent years, MFSS and similar lab-on-chip systems have gained increasing recognition in clinical embryology and are likely to play a crucial role in future sperm selection for ART. For a detailed evaluation of the effectiveness of separation techniques, it is necessary to assess effectiveness not only in terms of the impact on pregnancy rates, which is difficult to iterate given the large number of other influences. In our view, this approach needs to be complemented by a detailed evaluation of sperm, which should include not only standard evaluation parameters but also detailed analyses such as detection of DNA status, protamination, acrosome, or mitochondrial status. In the future, it will be appropriate to use this system to assess sperm in detail, not only in normozoospermic patients, but also in all patients. The ejaculate of patients with non-normozoospermia may not always be suitable for this type of separation. We found a positive effect on the DNA integrity status of sperm in this study; however, in patients with oligozoospermia with very low concentrations, this method may not be effective and may limit obtaining sufficient sperm. MFFS methods only work with a limited volume (<1.0 mL) of ejaculate and in patients who have a large volume and low concentration (<1.0 × 10^6^/mL), this approach may result in only a very small amount of sperm being recovered after separation, which may be limiting for the actual ICSI technique. Low sperm concentration and, ultimately, a low sperm count after MFSS sorting can lead to insufficient sperm for actual performance. For example, intrauterine insemination techniques or the classical IVF method need in the order of thousands of sperm for each fertilized oocyte. Even the ICSI method, which is much less demanding in terms of sperm count, may be limited by insufficient quantities of isolated sperms. In our study, we found that in patients with a concentration below 10 mil/mL and worse motility, there were more samples with less than 0.1 mil of separated sperm after the MFSS technique than after the swim-up technique. Indeed, in the case of the swim-up method, this risk is considerably lower because the system can be more easily optimized. It is possible to process a large volume of ejaculate, and at very low concentrations, it is possible to reduce the volume of the swim-up media and increase the concentration of sperm in the swim-up media, making it suitable for actual separation.

## 4. Materials and Methods

### 4.1. Study Design, Patient Eligibility Criteria and Sample Division

This non-invasive cross-sectional observation study was approved by the Ethics Committee of the University Hospital Brno on January 2023 in Brno, Czech Republic (reference number 11-110123/EK). In this study, we used leftover samples from routine spermiogram analysis. Men from infertile couples who underwent therapy at CERMED (Brno, Czech Republic) from February 2023 to April 2025 were included. Informed consent was obtained from all patients. The eligibility criteria were age 18–60 and a signed informed consent form. The inclusion criteria were a minimum volume of 1.6 mL and the presence of sperm in semen (azoospermic patients were excluded). A total of 68 patients from the CERMED center aged 21–47 years were included in this study. Semen specimens were obtained following an abstinence period of 2–5 days and were collected in sterile containers via masturbation. The liquefaction of the samples occurred at room temperature within 30 min. If the semen did not liquefy during the first 30 min, we waited another 30 min. We first homogenized the ejaculate thoroughly by repeated passaging through a syringe (6–10 times), then measured its total volume, evaluated its pH and took aliquots for sperm evaluation according to the World Health Organization manual [38]. To validate the homogenization process, we used triplicate aliquots for both concentration and motility analysis and performed several rounds of validation before actually evaluating the samples. For sperm evaluation, we used three aliquots (50 μL) for sperm concentration assessment and three aliquots (10 μL) for sperm motility assessment from different areas of the sample to reduce the risk of error. After counting, aliquots were used to evaluate sperm morphology and other sperm types. Sixty minutes after ejaculation, we took 20 µL of the sample for ROS evaluation.

Based on primary semen analysis, 40 patients were classified as normozoospermic, while 28 patients were identified as non-normozoospermic, according to the WHO manual [38]. For a patient to be classified as normozoospermic, the semen volume must be ≥1.5 mL, sperm concentration ≥15 × 10^6^/mL, total sperm count ≥39 × 10^6^/ejaculate, total motility ≥40%, progressive motility ≥32%, and normal morphology ≥4%. Patients with values lower than these criteria were classified as non-normozoospermic (oligozoospermic, low sperm concentration, or low total sperm count; asthenozoospermic, low total motility or low progressive motility; or teratozoospermic, low proportion of sperm with normal morphology). We used the WHO manual from 2010 [38] because our lab has successfully relied on it for many years, and this remained standard practice at the time we planned this study.

For each patient, the ejaculate was divided into three parts, 60 min after ejaculation, as follows:The first portion (180 µL) was not separated, only analyzed.The second portion (1000 µL) was separated using MFSS (Ca0 microfluidic chip; LensHOOKE Bonraybio, Taichung, Taiwan).The third portion (≥400 µL) was processed using the swim-up (SU) method (Figure 4).

### 4.2. Sperm Selection

#### 4.2.1. Microfluidic Sperm Sorting

Microfluidic sperm sorting was performed using a commercial Ca0 device (LensHOOKE, Bonraybio, Taichung, Taiwan), according to the manufacturer’s instructions. Briefly, 1000 μL of neat semen was introduced into the lower chamber and 900 μL of Sperm Preparation Medium (Origio, Ballerup, Denmark) was added to the upper chamber and covered with a cover piece. The device was placed in a humidified incubator at 37 °C for 30 min. After incubation, 500 μL of the sperm suspension was extracted from the upper chamber for subsequent analysis (Figure 4).

#### 4.2.2. Swim-Up Method

Semen samples were pipetted into 15 mL conical centrifuge tubes and washed twice in 2 mL Sperm Preparation Medium (REF 10700060A, Origio, Ballerup, Denmark) following the manufacturer’s protocol. After the second wash, 1000 μL of Universal IVF medium (REF 10310060A; Origio, Ballerup, Denmark) was gently overlaid onto the sperm suspension. The tube was inclined at approximately 45° to enhance the interface between the semen culture medium and subsequently incubated for 60 min at 37 °C. After incubation, 400 μL of the uppermost layer was collected for analysis.

### 4.3. Semen Analyses

Each patient underwent sperm analysis. The volume was determined using gravimetric measurements (analytic balance KERN ABJ320-4NM). Following the liquefaction of the ejaculate, a comprehensive sperm analysis was conducted, focusing on ejaculate volume, sperm concentration, progressive motility, percentage of sperm with normal morphology, and total sperm count (volume × sperm concentration). After separation, the following parameters were assessed: sperm concentration, motility, morphology, DNA integrity, acrosomal status, mitochondrial membrane potential (MMP).

Sperm motility serves as an indicator of sperm separation efficiency and is a valuable parameter for assessing the success of sperm sorting. Motile spermatozoa are viable and have a low damage ratio. Motility was assessed as the sum of progressive and non-progressive spermatozoa and expressed as a percentage. Progressive motility refers to sperm that moves actively in either a straight line or a wide circular path, regardless of speed. Non-progressive motility includes all other movement patterns that do not result in a forward progression.

To evaluate sperm morphology, Diff-Quik staining (MICROPTIC SL Co., Barcelona, Spain) was performed [39]. A 10 μL sperm sample was spread onto a clean glass slide, air-dried at room temperature for 30 min, and stained following the manufacturer’s protocol. Bright-field microscopy was used to examine 200 spermatozoa per sample, categorizing them based on head, midpiece, and tail abnormalities. The proportion of normal spermatozoa was calculated and expressed as a percentage.

### 4.4. Assessment of DNA Fragmentation

Sperm DNA fragmentation was assessed using a sperm chromatin dispersion test Halosperm G2 kit (Halotech, Madrid, Spain), following the manufacturer’s protocol. Large and medium halos indicated non-fragmented sperm, while small halos, no halos, or degeneration classified sperm as fragmented. The DNA fragmentation index (DFI) was calculated using the following formula:$$DFI(\%)=\left(\frac{fragmented\;spermatozoa+degenerated\;spermatozoa}{total\;spermatozoa\;counted}\right)\times 100$$

A minimum of 600 spermatozoa per sample was evaluated using a 100× objective microscope, with at least 300 spermatozoa analyzed by two independent technicians to minimize bias. The DFI value of unprocessed samples served as a baseline for evaluating separation efficiency.

### 4.5. Evaluation of Acrosomal Status

Lectin peanut agglutinine labeled fluorescein isothiocyanate (FITC-PNA) was used to assess the acrosomal condition of sperm cells, according to the method stated by Estevez and Verza, 2011 [40]. Sperm samples (5 μL) were air-dried on slides, fixed in ice-cold ethanol, coated with 40 µg/mL FITC-PNA (Sigma-Aldrich, Steinheim, Germany) in PBS, and incubated in the dark for 30 min. The glass was then washed with PBS, and the samples were mounted with Vectashield medium containing DAPI (Vector Laboratories, Newark, NJ, USA). After thorough washing with PBS (Sigma Aldrich, Steinheim, Germany), the coverslips were applied. A fluorescence microscope (Nikon ECLIPSE Ts2, Nikon, Tokyo, Japan) was used to examine the fluorescence patterns of 200 sperm cells per sample from random areas at 1000x magnification. Acrosome status was classified as follows: (1) complete acrosome staining, indicating an intact acrosome; (2) partial acrosome staining, signifying sperm with disturbed acrosome; or (3) no staining of the entire sperm head, indicating a sperm with a detached acrosome (Figure 2). The percentage of staining patterns was evaluated [41].

### 4.6. Mitochondrial Evaluation

Mitochondrial membrane potential (MMP) was assessed using JC-1, a lipophilic cationic dye [42]. Following the manufacturer’s instructions (cat. #10009172; Cayman Chemical Company, Ann Arbor, MI, USA), a JC-1 working solution was prepared by diluting the stock solution 1:10 (*v*/*v*) in the culture medium. JC-1 (1 μL) was mixed with 9 μL of the sperm suspension and incubated for 30 min in a humidified incubator (37 °C, 5% CO_2_, dark conditions). Mitochondrial activity was examined under a fluorescence microscope (Nikon ECLIPSE Ts2, Tokyo, Japan) at 1000× magnification. Sperm with high MMP exhibited orange-red fluorescence, and the results are expressed as the percentage of sperm displaying high MMP.

### 4.7. ROS/RNS Assessment

The concentration of reactive oxygen/nitrogen species (ROS/RNS, including hydrogen peroxide as one of the primary reactive oxygen species present in semen) in semen samples was measured using an OxiSelect™ In Vitro ROS/RNS Assay Kit (Cell Biolabs, San Diego, CA, USA) following the manufacturer’s instructions. Fluorescence was detected using a Fluostar Omega Microplate Reader (BMG Labtech, Ortenberg, Germany) at an excitation wavelength of 485/20 nm and emission wavelength of 528/20 nm. Because ROS/RNS levels decrease over time, samples were frozen immediately after liquefaction in liquid nitrogen (−196 °C). Storage at −20 °C resulted in a rapid decline in ROS/RNS (Appendix A).

### 4.8. Statistical Evaluation

Statistical analyses were performed using STATISTICA CZ version 10 (StatSoft, Prague, Czech Republic). Data are presented as the mean ± standard deviation. Group comparisons were conducted using one-way analysis of variance (ANOVA). Statistical significance was set at *p* < 0.05.

## 5. Conclusions

This study compared the swim-up method and the microfluidic sperm sorting (MFSS) system in patients with different spermiogram parameters. Our findings indicated that in normozoospermic patients, no significant differences were observed between the two methods. However, in non-normozoospermic patients, MFSS demonstrated a significant reduction in sperm DNA fragmentation, suggesting a potential benefit in these cases. This study compared the swim-up method and microfluidic sperm selection sorting system (MFSS) in patients with different spermiogram parameters. Our findings indicate that in normozoospermic patients, no significant differences were observed between the two methods. In non-normozoospermic patients, MFSS demonstrated a significant reduction in sperm DNA fragmentation, suggesting a potential benefit in these cases. However, this method is not suitable for all patients with non-normozoospermia. In patients with low concentration and poor motility, the use of MFSS may lead to a very low number of separated sperm, which may limit the final selection of sperm for ICSI. Although microfluidic selection systems are increasingly being recognized as the future of sperm selection in assisted reproduction, their development has remained stagnant for nearly two decades. Nevertheless, in recent years, MFSS and other lab-on-chip technologies have gained attention in clinical embryology, positioning them as promising advancements in sperm selection for IVF.

## Figures and Tables

**Figure 1 ijms-26-05374-f001:**
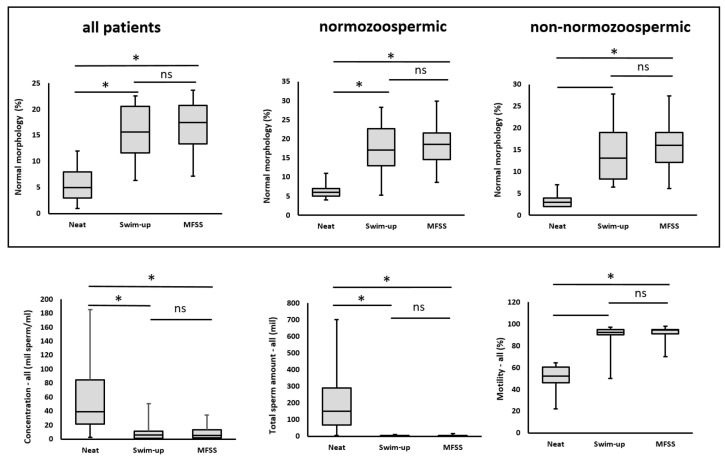
Results of swim-up and MFSS separation techniques in comparison to neat ejaculate. Box plots illustrate the data distribution (ns, non significant; * *p*  <  0.05).

**Figure 2 ijms-26-05374-f002:**
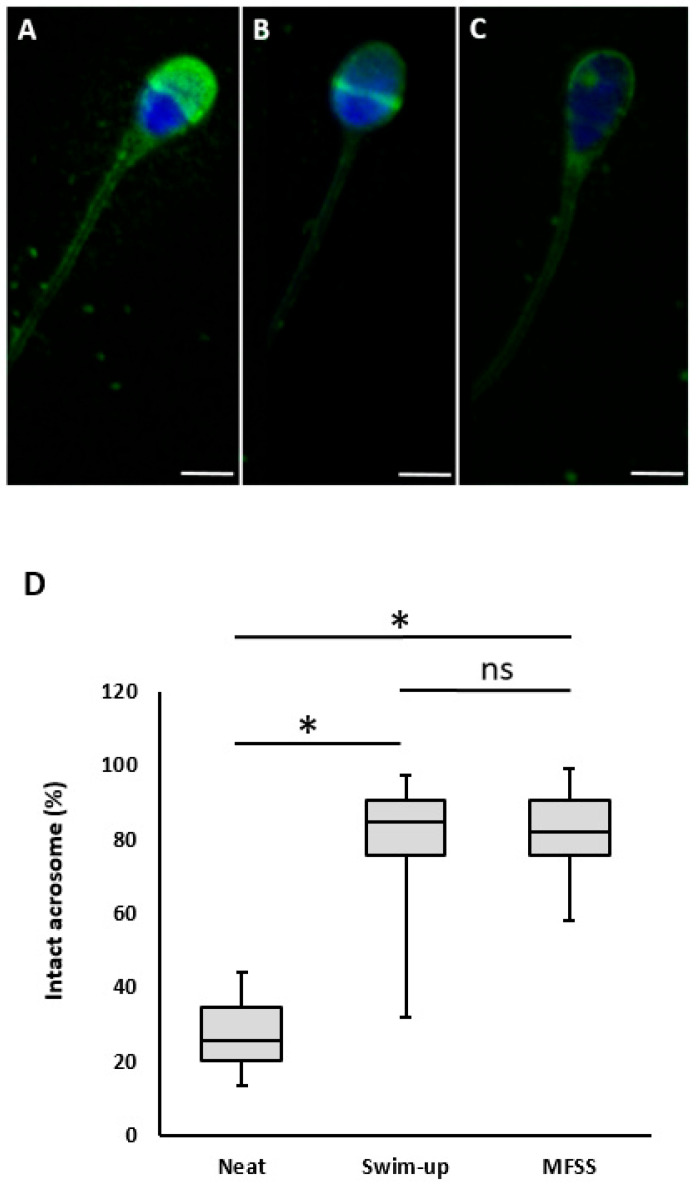
Evaluation of acrosomal status. Spermatozoa acrosome labeled by FITC-PNA: (**A**) intact acrosome, (**B**) equatorial region labelling, (**C**) non-detected acrosome; green, acrosome; blue, DNA. The scale bar represents 5 µm. (**D**) Box plot illustrates the data distribution (ns, non-significant; * *p*  <  0.05).

**Figure 3 ijms-26-05374-f003:**
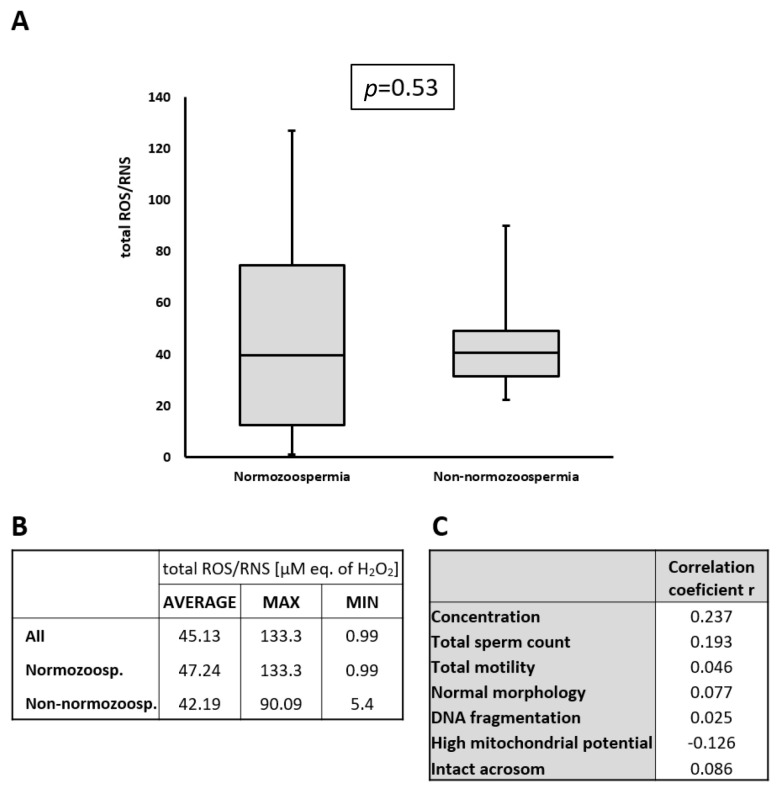
ROS/RNS values in seminal plasma. (**A**) graphical presentation of ROS/RNS values in normozoospermic and non-normozoospermic patients. The 25th and 75th percentiles are represented by boxes, with the median values, while the 10th and 90th percentiles are represented by whiskers; (**B**) value of ROS/RNS; (**C**) correlation between ROS/RNS H_2_O_2_ concentration and sperm parameters.

**Figure 4 ijms-26-05374-f004:**
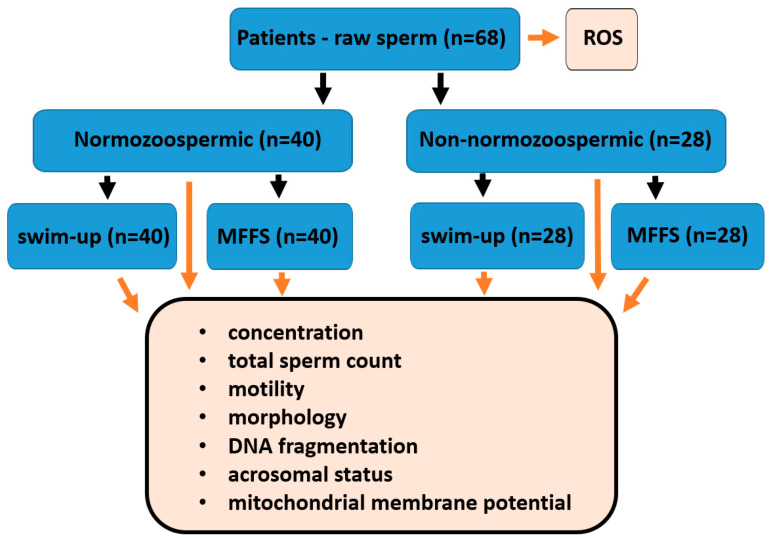
Flow diagram of the study design and analysis: Schematic representation of the workflow. Fresh ejaculate samples were collected 60 min post-ejaculation for assessment of initial reactive oxygen species (ROS) levels. Only samples with a minimum ejaculate volume of 1.6 mL were included. Based on standard sperm analysis, the patients were classified as normozoospermic or non-normozoospermic. Each ejaculate was divided into three portions, as follows: (I) an unprocessed control sample, (II) a sample separated using the swim-up (SU) method, and (III) a sample separated using the microfluidic sperm sorting system (MFSS). Following separation, sperm quality was assessed by evaluating the concentration, motility, morphology, DNA fragmentation index (DFI), mitochondrial membrane potential (MMP), and acrosomal status.

**Table 1 ijms-26-05374-t001:** Patient parameters.

Parameters	Mean ± SD
Number of patients	68
Age (years)	32.14 ± 7.1
Volume (mL)	3.93 ± 1.83
Concentration (×10^6^ cell/mL)	44.57 ± 31.12
Progressive motility (%)	43.91 ± 14.55
Total motility (%)	53.05 ± 13.81
Normal morphology (%)	5.3 ± 2.45
High MMP (%)	87.04 ± 3.92
DFI (%)	18.27 ± 8.36
Intact acrosome (%)	29.15 ± 8.77
ROS (µM, H_2_O_2_)	45.13 ± 32.19
Normozoospermic patients	n = 40
Non-normozoospermic patients	n = 28

DFI, DNA fragmentation index; MMP, mitochondrial membrane potential; ROS, reactive oxygen species.

**Table 2 ijms-26-05374-t002:** Sperm parameters of all patients before and after separation.

Parameters	Ejaculate	Separation Methods	Mean ± SD
Concentration (×10^6^ sperm/mL)	44.57 ± 31.12	Swim-up	7.68 ± 7.75 ^a^
		MFSS	8.52 ±8.69 ^a^
Total sperm count (×10^6^)	177.87 ± 141.6	Swim-up	3.72 ± 4.25 ^a^
		MFSS	3.96 ± 4.01 ^a^
Total motility (%)	53.05 ± 13.8	Swim-up	91.29 ± 8.72 ^a^
		MFSS	93.57 ± 4.24 ^a^
Normal morphology (%)	5.3 ± 2.45	Swim-up	16.01 ± 6.21 ^a^
		MFSS	17.26 ± 5.34 ^a^
DNA fragmentation (%)	18.27 ± 8.36	Swim-up	9.36 ± 7.5 ^a^
		MFSS	5.98 ± 5.95 ^b^
High mitochondrial potential (%)	87.04 ± 3.85	Swim-up	92.85 ± 2.43 ^a^
		MFSS	92.79 ± 2.75 ^a^
Intact acrosome (%)	29.15 ± 8.77	Swim-up	81.09 ± 13.61 ^a^
		MFSS	81.78 ± 10.6 ^a^

Values with different superscripts (a, b) are significantly different (*p* ˂ 0.05).

**Table 3 ijms-26-05374-t003:** Evaluated parameters in normozoospermic patients after separation.

Parameters	Swim-Up	MFSS	*p*-Value	Neat
Concentration (×10^6^ sperm/mL)	9.6 ± 6.84	11.53 ± 9.02	0,28	55.25 ± 29.32
Total sperm count (×10^6^)	4.44 ± 3.21	5.36 ± 4.2	0,27	202.34 ± 107.46
Total motility (%)	92.5 ± 5.16	93.06 ± 4.48	0,39	58.95 ± 10.4
Normal morphology (%)	16.42 ± 5.99	16.50 ± 6.1	0,95	6.62 ± 1.99
DNA fragmentation (DFI %)	9.27 ± 7.02	6.69 ± 6.36	0,08	17.6 ± 8.76
High mitochondrial potential (%)	93.05 ± 2.85	93.62 ± 2.8	0,37	88.24 ± 4.22
Intact acrosome (%)	85.09 ± 12.27	85.94 ± 10.16	0,73	28.82 ± 8.07

**Table 4 ijms-26-05374-t004:** Evaluated parameters in non-normozoospermic patients after separation.

Parameters	Swim-Up	MFSS	*p*-Value	Neat
Concentration (×10^6^ sperm/mL)	3.98 ± 5.63	4.3 ± 5.59	0.83	28.89 ± 27.43
Total sperm count (×10^6^)	1.87 ± 2.7	2.01 ± 2.6	0.84	113.08 ± 148.24
Total motility (%)	89.57 ± 12.05	93.78 ± 3.94	0.18	44.64 ± 13.86
Normal morphology (%)	13.22 ± 5.95	14.32 ± 5.7	0.48	3.42 ± 1.72
DNA fragmentation (%)	10.0 ± 7.83	5.69 ± 6.32	0.027	19.24 ± 7.81
High mitochondrial potential (%)	93.42 ± 2.56	92.67 ± 3.28	0.34	86.73 ± 4.09
Intact acrosome (%)	80.79 ± 14.13	81.25 ± 10.8	0.89	26.45 ± 7.34

## Data Availability

The data and material used in this research are available from the corresponding author upon request.

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
