# Peer review of "Comparison of Swim-Up and Microfluidic Sperm Sorting Methods in Selection of Sperm for Intracytoplasmic Sperm Injection"

_ijms, 2025, doi:10.3390/ijms26115374_

Round 1

Reviewer 1 Report (New Reviewer)

Comments and Suggestions for Authors

The paper requires substantial corrections before publishing.

L. 17: "MFSS" stands for "microfluidic sperm sorting", not "microfluidic sperm selection systems"

L. 18: "increasing due to" instead of "increasing to"

L. 19: in English "sperm analysis" is more common used instead of "spermiogram"

L. 23: "DNA integrity" and "DNA fragmentation" are used promiscue, however, these terms are opposites

L. 25: "an analysis of parameters between separation techniques" does not make any sense

L. 49+52: "cervix uteri" and "in vitro" should be written in italic

L. 64: "This method" - which one? in the previous sentence you are mentioning 2 different methods

L. 65+67: Both sentences are beginning with "However"

L. 65: "these methods are not perfect and centrifugation, has been" - redundant comma

L. 87: "comparison of a microfluidic sperm selection system using the swim-up method" - maybe "... system AND swim-up"?

L. 99: "assess" seems to be more appropriate than "minimize"

Methodology should be stated between "Introduction" and "Results"

L. 95: "A total of 68..." - there is a verb missing in the sentence

Table 1 is named "A descriptive evaluation of spermatozoa", however, number of patients, age, volume, normozoospermic patients and non-normozoospermic patients are not characteristics of spermatozoa

L. 107: "therefore in yielding sufficient" - the sentence does not make any sense

L. 113: Why is total sperm count not in the Table 1?

L. 122+123: "High motility confirms the effectiveness of the separation process, ensuring that only viable, undamaged sperm are included" - that's not true, high motility is not a guarantee for high-quality sperm

L. 143: "considered one" -> "considered as one"

L. 148-151: The two sentences have the same meaning!

I don't understand the difference between table 1 and 2 - both should describe semen parameters before separation?

In tables 1 and 2 you are using "mL", but in tables 3 and 4 you are using "ml" - it must be unified

Table 2: mean sperm concentration and SD is totally same for Swim-up and MFSS?

L. 155: "comparison... in compare"?

L. 155: "technics" - "techniques"

L. 153: "superscripts (A, B)", but in the table superscripts "a" and "b" are used

Fig. 2: What is "Ca0"?

L. 166: "Box plot illustrate" -> "illustrates"

L. 169: "status a following" - redundant "a"

L. 172: MFSS - see above

L. 173: "non-selected group" - is it not rather a "sperm before separation"?

L. 181: ROS - the abbreviation has already been explained

L. 194: "25th and 75th" -> "25th and 75th"

L. 196: "value" -> "values"

L. 201: "(n=28), That" -> "(n=28). That"

L. 203: "patient" -> "patients"

Table 3: "106" -> "106"

Table 3: "Total sperm count" instead of "Total amount of sperm" should be used

L. 220: "observed" -> "evaluated"

L. 222: "both" -> "each" ("both" means, that both methods were perfomed consecutively in all samples)

L. 231+232: "in the proportion" -> "in terms of proportion"

L. 273: "to133" -> "to 133"

L. 281: "DNA fragmentation, in non-normozoospermic" - redundant comma

L. 285: "patient characteristics" -> "patient's characteristics"

L. 287: "its development" -> "whose development"

L. 296: "not only in normozoospermic patients" - should it not rather be "not only in non-normozoospermic patients"?

Fig. 4: "Total sperm count" instead of "Total amount of sperm" should be used

L. 363: The heading is "DNA integrity", however, you are describing DNA fragmentation, which is opposite

L. 377: "Adrich" -> "Aldrich"

L. 382: "×1000" -> "1000×"

L. 396: "the percentage of staining patterns among was recorded" - among what?

L. 410: "P < 0.05" -> "p < 0.05"

Table S1: "mounths" -> "months"

Comments on the Quality of English Language

Since some phrases are hard to understand, a proof-reading by an English native speaker must be performed.

Author Response

Thank you very much for your time, here are our reactions:

17: "MFSS" stands for "microfluidic sperm sorting", not "microfluidic sperm selection systems"

We agree it was changed.

18: "increasing due to" instead of "increasing to" We agree it was changed.

19: in English "sperm analysis" is more common used instead of "spermiogram"

We agree with this comment. Abstract was rewritten

23: "DNA integrity" and "DNA fragmentation" are used promiscue, however, these terms are opposites. Now is it correct.

25: "an analysis of parameters between separation techniques" does not make any sense We agree it was rewritten.

49+52: "cervix uteri" and "in vitro" should be written in italic Corrected

64: "This method" - which one? in the previous sentence you are mentioning 2 different methods We agree wirth this point, it was rewritten.

65+67: Both sentences are beginning with "However" We agree it was rewritten.

65: "these methods are not perfect and centrifugation, has been" - redundant comma We agree it was rewritten.

87: "comparison of a microfluidic sperm selection system using the swim-up method" - maybe "... system AND swim-up"?

99: "assess" seems to be more appropriate than "minimize" We agree it was rewritten.

Methodology should be stated between "Introduction" and "Results"

Thank you for pointing this out, but this is not possible. After my asking to editor he recommend me keep this format which follows to Journal's Regulations.

  1. 95: "A total of 68..." - there is a verb missing in the sentence We agree it was changed.

Table 1 is named "A descriptive evaluation of spermatozoa", however, number of patients, age, volume, normozoospermic patients and non-normozoospermic patients are not characteristics of spermatozoa

It was corrected

107: "therefore in yielding sufficient" - the sentence does not make any sense. It was corrected

113: Why is total sperm count not in the Table 1? This parameter is in table 2, in table are parametrs about patients and main sperm parameters.

122+123: "High motility confirms the effectiveness of the separation process, ensuring that only viable, undamaged sperm are included" - that's not true, high motility is not a guarantee for high-quality sperm. Is it true now is it corrected.

143: "considered one" -> "considered as one" Corrected

148-151: The two sentences have the same meaning! Corrected

I don't understand the difference between table 1 and 2 - both should describe semen parameters before separation?

In table 1 are parameters of patients with main sperm parameters. In table 2 are results before and after swim up and after MFSS separation.

In tables 1 and 2 you are using "mL", but in tables 3 and 4 you are using "ml" - it must be unified. Now is it correct.

Table 2: mean sperm concentration and SD is totally same for Swim-up and MFSS? It was corrected my mistake

155: "comparison... in compare"? Corrected

155: "technics" - "techniques" Corrected

153: "superscripts (A, B)", but in the table superscripts "a" and "b" are used. Is it corrected.

Fig. 2: What is "Ca0"? We agree, Figure was changed, legend was changed.

166: "Box plot illustrate" -> "illustrates"  Corrected

169: "status a following" - redundant "a" Corrected

172: MFSS - see above Corrected

173: "non-selected group" - is it not rather a "sperm before separation"? Corrected

181: ROS - the abbreviation has already been explained Corrected

194: "25th and 75th" -> "25th and 75th" Corrected

196: "value" -> "values"  Corrected

201: "(n=28), That" -> "(n=28). That" Corrected

203: "patient" -> "patients"           Corrected

Table 3: "106" -> "106"

Table 3: "Total sperm count" instead of "Total amount of sperm" should be used

220: "observed" -> "evaluated" Corrected

222: "both" -> "each" ("both" means, that both methods were perfomed consecutively in all samples)

231+232: "in the proportion" -> "in terms of proportion"

273: "to133" -> "to 133" Corrected

281: "DNA fragmentation, in non-normozoospermic" - redundant comma Corrected

285: "patient characteristics" -> "patient's characteristics"  Corrected

287: "its development" -> "whose development"   Corrected

296: "not only in normozoospermic patients" - should it not rather be "not only in non-normozoospermic patients"?

Corrected

Fig. 4: "Total sperm count" instead of "Total amount of sperm" should be used  Corrected

363: The heading is "DNA integrity", however, you are describing DNA fragmentation, which is opposite Corrected

377: "Adrich" -> "Aldrich" Thank you for pointing this out, now is it correct. 

382: "×1000" -> "1000×" Thank you for pointing this out, now is it correct. 

396: "the percentage of staining patterns among was recorded" - among what? Corrected

410: "P < 0.05" -> "p < 0.05" Corrected

Table S1: "mounths" -> "months" Corrected

Reviewer 2 Report (New Reviewer)

Comments and Suggestions for Authors

Thank you for the opportunity to review this manuscript. Overall, I find this study highly valuable. By comparing swim-up and microfluidic sperm selection systems using split-sample analysis, the authors provide meaningful insights into sperm quality and quantity as assessed by each method. I believe these findings have the potential to advance treatment strategies in reproductive medicine. The manuscript has strong potential for acceptance, provided that the authors address several points requiring clarification.

Major Points

  1. Justification for Assuming the Ejaculate is Uniform
    The study design relies on the assumption that the ejaculate is a completely uniform liquid, as evidenced by splitting the sample into a first portion (200 μL), second portion (1000 μL), and third portion (≥500 μL). Ideally, to justify this assumption, the authors would conduct a preliminary verification that the quantitative and qualitative parameters of sperm prior to separation are the same among the first, second, and third portions. Is there any such experimental result available? Alternatively, have the authors referred to previous studies on methods that ensure complete homogenization of semen?

    • Rationale for the Question: Semen is composed of a viscous, gel-like fraction from the seminal vesicles and a more watery fraction from the prostate, leading to a potentially heterogeneous fluid. Complete mixing may be difficult (PMID: 14656406). In practice, sperm in prostatic fluid has been reported to have a lower DNA fragmentation rate than sperm in fluid from the seminal vesicles (PMID: 25547665). Furthermore, the sperm concentration in the prostatic fluid portion can differ significantly from that in the seminal vesicle portion (PMID: 38807752). Thus, dividing semen into three portions does not necessarily result in an accurate three-way partitioning of the sperm themselves.

  2. Reconciliation of the Decreased Sperm Yield via MFSS with the Main Outcome
    In the Discussion, the authors note that the decrease in sperm count after separation is a key limitation of MFSS, and they further state that while this is not problematic for normozoospermic patients, it could be a concern for those with low semen volume or sperm concentration. However, the main outcome of this study shows that MFSS did not offer a clear advantage over swim-up in normozoospermic patients, whereas in non-normozoospermic patients, MFSS achieved lower DNA fragmentation compared to swim-up. This might seem partially contradictory, given that the Discussion suggests a limitation of MFSS specifically for low sperm counts. The authors note that “The choice of sperm separation method should be individualized based on the patient characteristics.” To resolve this apparent contradiction, it would be helpful to include additional explanation or discussion, clarifying how MFSS can still be beneficial for certain non-normozoospermic men (e.g., those with higher sperm DNA fragmentation), despite the reduced sperm yield.

  3. Completeness of Reporting According to Standard Guidelines
    Some aspects of the study design typically required for observational research are not fully described. Please consider whether the STROBE Statement would be applicable, and if so, add any relevant items to enhance clarity.

Minor Points

  1. Distinction Between “Sperm Sorting” and “Sperm Selection”
    The manuscript repeatedly uses both terms—“sperm sorting” and “sperm selection.” Please clarify whether these are intended to be synonymous or if there is a specific difference in how these terms are applied.

  2. Details on the Source of Semen Samples
    It remains unclear how the semen samples were obtained:

    • Were these leftover samples from routine clinical semen analysis?

    • Or were they newly collected specifically for research from volunteers?
      Please clarify this in the manuscript.

  3. Study Design Description
    In the Materials and Methods section, explicitly state the study design. If leftover samples from routine testing were used, this might be a non-invasive, cross-sectional observational study. If new samples were collected specifically for this research, it would be an invasive observational study, and prospective registration of the study design might be recommended.

  4. Description of “Patients”
    Please clarify the population from which you recruited participants. For instance, were they healthy spouses of infertile women seeking care at University Hospital Brno, or men presenting with male infertility? If helpful, refer to the STROBE Statement and include the relevant eligibility criteria.

  5. Definition of Normozoospermic vs. Non-Normozoospermic
    You mention, “Based on primary semen analysis 30 patients were classified, according to the WHO manual [38].” The cited reference (WHO 2010, 5th edition) includes multiple parameters (semen volume, sperm concentration, total sperm count, motility, progressive motility, morphology, vitality, white blood cell count, etc.) and different WHO editions present different reference ranges.

    • Please list the exact cutoff values you used to classify participants as normozoospermic or non-normozoospermic. For example, if you set “semen volume ≥1.5 mL,” “sperm concentration ≥15 million/mL,” “total sperm count ≥39 million/ejaculate,” “motility ≥40%,” “progressive motility ≥32%,” “morphology ≥4%,” etc., then explicitly mention these criteria so readers fully understand your classification method.

  6. Rationale for Using the 2010 WHO Manual (5th Edition)
    The WHO manual for human semen analysis was updated in 2021 (6th edition). Please explain why you chose to use the older 2010 reference. For instance, was it due to your laboratory protocols, or was the study planned and initiated before the newest edition was published?

  7. Placement of Certain Explanatory Text
    In section “2.2.3 Sperm motility,” there is an explanation of the definitions of “progressive” and “non-progressive” motility. If the Materials and Methods section already provides sufficient definitions, repeating them in the Results may be unnecessary. According to the STROBE Statement, the Results section is typically structured into participant flow, descriptive data, outcome data, main results, and other analyses. Explanatory or definitional text is best kept in the Materials and Methods unless necessary for clarity. Consider whether the length of some explanations in the Results can be reduced to improve readability.

Author Response

Thank you for your time our answers are below:

Major Points

Justification for Assuming the Ejaculate is Uniform

The study design relies on the assumption that the ejaculate is a completely uniform liquid, as evidenced by splitting the sample into a first portion (200 μL), second portion (1000 μL), and third portion (≥500 μL). Ideally, to justify this assumption, the authors would conduct a preliminary verification that the quantitative and qualitative parameters of sperm prior to separation are the same among the first, second, and third portions. Is there any such experimental result available? Alternatively, have the authors referred to previous studies on methods that ensure complete homogenization of semen?

Thank you for this point. In text was not mentioned homogenization as very important step of sperm analyses. Is it necessary to make thorouh homogenization of sample before any sperm analyses (according to WHO manual) and is it same in this study. For erification of homogenization proces we did 3 aliquots  for concentration and 3 for motility analyses. We did several times verification of our homogenization steps before analyses. It works well and it was also several times presented (WHO manual 2010) After this homogenization we took part for analyses and after final analyses we will take samples for sorting. Detailed about sample preparation are now in text

Rationale for the Question: Semen is composed of a viscous, gel-like fraction from the seminal vesicles and a more watery fraction from the prostate, leading to a potentially heterogeneous fluid. Complete mixing may be difficult (PMID: 14656406). In practice, sperm in prostatic fluid has been reported to have a lower DNA fragmentation rate than sperm in fluid from the seminal vesicles (PMID: 25547665). Furthermore, the sperm concentration in the prostatic fluid portion can differ significantly from that in the seminal vesicle portion (PMID: 38807752). Thus, dividing semen into three portions does not necessarily result in an accurate three-way partitioning of the sperm themselves.

This is absolutely true and it is necessary to be very careful when evaluating the ejaculate. In our case, first the ejaculate is completely homogenized, its total volume is determined, pH is determined, a small sample is taken for ROS assay and aliquots are taken for sperm evaluation according to WHO. Only then are the samples prepared for separation.

Reconciliation of the Decreased Sperm Yield via MFSS with the Main Outcome

In the Discussion, the authors note that the decrease in sperm count after separation is a key limitation of MFSS, and they further state that while this is not problematic for normozoospermic patients, it could be a concern for those with low semen volume or sperm concentration. However, the main outcome of this study shows that MFSS did not offer a clear advantage over swim-up in normozoospermic patients, whereas in non-normozoospermic patients, MFSS achieved lower DNA fragmentation compared to swim-up. This might seem partially contradictory, given that the Discussion suggests a limitation of MFSS specifically for low sperm counts. The authors note that “The choice of sperm separation method should be individualized based on the patient characteristics.” To resolve this apparent contradiction, it would be helpful to include additional explanation or discussion, clarifying how MFSS can still be beneficial for certain non-normozoospermic men (e.g., those with higher sperm DNA fragmentation), despite the reduced sperm yield.

you are right that it is not sufficiently justified. The text has now been added

Completeness of Reporting According to Standard Guidelines

Some aspects of the study design typically required for observational research are not fully described. Please consider whether the STROBE Statement would be applicable, and if so, add any relevant items to enhance clarity.

STROBE manual for observational studies in epidemiology. This is not quite the case, for the verification of our hypothesis the patient's identity is not so important, but the effectiveness of the separation system in all patients from normal to infertile. However, a part of the text related to patients has been added.

Minor Points

Distinction Between “Sperm Sorting” and “Sperm Selection”

The manuscript repeatedly uses both terms—“sperm sorting” and “sperm selection.” Please clarify whether these are intended to be synonymous or if there is a specific difference in how these terms are applied.

Thank you, now is it corrected, sorting is used in case of MFFS and sorting methods, selection is term for general slectio of sperm – thermotaxis etc.

Details on the Source of Semen Samples

It remains unclear how the semen samples were obtained:

Were these leftover samples from routine clinical semen analysis?

Yes is it leftover after spermiogram analyses.

Or were they newly collected specifically for research from volunteers? no

Please clarify this in the manuscript.

Study Design Description

In the Materials and Methods section, explicitly state the study design. If leftover samples from routine testing were used, this might be a non-invasive, cross-sectional observational study. If new samples were collected specifically for this research, it would be an invasive observational study, and prospective registration of the study design might be recommended.

We agree, now is this mention included in 4.1.part.

Description of “Patients”

Please clarify the population from which you recruited participants. For instance, were they healthy spouses of infertile women seeking care at University Hospital Brno, or men presenting with male infertility? If helpful, refer to the STROBE Statement and include the relevant eligibility criteria.

It was corrected and defined inclusion criteria.

Definition of Normozoospermic vs. Non-Normozoospermic

You mention, “Based on primary semen analysis 30 patients were classified, according to the WHO manual [38].” The cited reference (WHO 2010, 5th edition) includes multiple parameters (semen volume, sperm concentration, total sperm count, motility, progressive motility, morphology, vitality, white blood cell count, etc.) and different WHO editions present different reference ranges.

Please list the exact cutoff values you used to classify participants as normozoospermic or non-normozoospermic. For example, if you set “semen volume ≥1.5 mL,” “sperm concentration ≥15 million/mL,” “total sperm count ≥39 million/ejaculate,” “motility ≥40%,” “progressive motility ≥32%,” “morphology ≥4%,” etc., then explicitly mention these criteria so readers fully understand your classification method.

OK we agree now is changed in section 4.1.

Rationale for Using the 2010 WHO Manual (5th Edition)

The WHO manual for human semen analysis was updated in 2021 (6th edition). Please explain why you chose to use the older 2010 reference. For instance, was it due to your laboratory protocols, or was the study planned and initiated before the newest edition was published?

For this study, the old WHO standard from 2010 was used, which does not differ dramatically from the new WHO2021 and was successfully used in our workplace for a long time and is popular in doctors of our hospital. At the time of planning and preparation of this study, the WHO standard from 2010 was commonly used, and it is still used in many workplaces today.

Placement of Certain Explanatory Text

In section “2.2.3 Sperm motility,” there is an explanation of the definitions of “progressive” and “non-progressive” motility. If the Materials and Methods section already provides sufficient definitions, repeating them in the Results may be unnecessary. According to the STROBE Statement, the Results section is typically structured into participant flow, descriptive data, outcome data, main results, and other analyses. Explanatory or definitional text is best kept in the Materials and Methods unless necessary for clarity. Consider whether the length of some explanations in the Results can be reduced to improve readability.

This part has been edited and moved to the materials and methods section

Reviewer 3 Report (New Reviewer)

Comments and Suggestions for Authors

Dear Authors,
In the current study, “Comparison of swim-up and microfluidic sperm sorting methods in selection of sperm for intracytoplasmic sperm injection” the authors demonstrated the topic is current and relevant, as it explores the effectiveness of microfluidic sperm selection systems for the treatment of infertility, with an emphasis on reducing DNA fragmentation in patients with impaired sperm parameters.
Some suggestions:
1.    In the introduction, you should provide more information on the real effectiveness of microfluidic sperm selection systems (MFSS) compared to the standard swim-up method, and what are their applicability and benefits in patients with impaired sperm parameters?
2.     In sections 2.2.2 and 2.2.3 (total sperm count and motility), although a significant decrease in total sperm count after selection is reported, the text lacks a more detailed analysis of the clinical significance of this decrease, especially for patients with severe oligozoospermia or asthenozoospermia. There is no specific assessment of whether the decreased count is sufficient for successful ICSI, especially with extremely low initial counts.
3.    In the results section:
- It is necessary to add a comment or analysis that considers the clinical applicability of the reduced sperm count after selection in patients with severely impaired sperm parameters.
-It should be considered whether in these cases MFSS is advisable to use as a first-choice method over SU, due to lower DNA fragmentation and preservation of sperm functional qualities, as well as to specify a threshold value for the minimum number of sperm required for ICSI after selection with both methods.
4.    In the discussion section, you should specifically comment on the results:
- How might limiting the volume of ejaculate processed to 1.0 mL affect the effectiveness of MFSS in patients with low ejaculate volume or low sperm concentration?
- What is the impact of low sperm yield after MFSS selection on the ability to obtain sufficient sperm for subsequent procedures, especially in patients with low sperm parameters?
- What is the risk of additional sperm damage in patients with high levels of reactive oxygen species (ROS) if MFSS does not provide rapid separation of sperm from seminal plasma?
- There is no clear distinction between summary of results and their interpretation, and the transitions between topics (e.g. ROS, MMP, DFI) are weak and abruptly change focus?
5.    The discussion should emphasize what exactly is the new contribution of this research.
6.    In the methods section:
- Why did the study only include patients with ejaculate volumes above 1.6 mL, which limits the applicability of the results to men with low ejaculate volumes and does this not compromise the assessment of the effectiveness of MFSS in this group?
7.    Conclusion:
It is not stated what are the limitations of the MFSS method in patients with lower ejaculate volume or high levels of reactive oxygen species (ROS) and how this method can be optimized for these patient groups?
The limitations of the current study need to be determined.

The English could be improved to more clearly express the research. 
Best Regards!

Comments on the Quality of English Language

The English could be improved to more clearly express the research. 

Author Response

Dear reviewer

Thank you very much for your time and are responces are below:

Some suggestions:

  1. In the introduction, you should provide more information on the real effectiveness of microfluidic sperm selection systems (MFSS) compared to the standard swim-up method, and what are their applicability and benefits in patients with impaired sperm parameters?

The introduction was supplemented -

  1. In sections 2.2.2 and 2.2.3 (total sperm count and motility), although a significant decrease in total sperm count after selection is reported, the text lacks a more detailed analysis of the clinical significance of this decrease, especially for patients with severe oligozoospermia or asthenozoospermia. There is no specific assessment of whether the decreased count is sufficient for successful ICSI, especially with extremely low initial counts.

Detailed information was added into results and new mention is now in conclusion a discussion.

  1. In the results section:

- It is necessary to add a comment or analysis that considers the clinical applicability of the reduced sperm count after selection in patients with severely impaired sperm parameters.

A new paragraph has been added to the results and a new supplement section has been created with data focused on this (SupplS3)

-It should be considered whether in these cases MFSS is advisable to use as a first-choice method over SU, due to lower DNA fragmentation and preservation of sperm functional qualities, as well as to specify a threshold value for the minimum number of sperm required for ICSI after selection with both methods.

Threshold in MFSS method is described in text, threshold for swim-up is not applicable and this method can be modified for very low sperm count

  1. In the discussion section, you should specifically comment on the results:

- How might limiting the volume of ejaculate processed to 1.0 mL affect the effectiveness of MFSS in patients with low ejaculate volume or low sperm concentration?

It is limiting for some art methods, it is now described in the text

- What is the impact of low sperm yield after MFSS selection on the ability to obtain sufficient sperm for subsequent procedures, especially in patients with low sperm parameters?

this mainly concerns patients with low concentration, this is now explained in the text

- What is the risk of additional sperm damage in patients with high levels of reactive oxygen species (ROS) if MFSS does not provide rapid separation of sperm from seminal plasma?

It is unclear whether rapid separation of sperm from seminal plasma has a positive impact, so this part was omitted from the discussion.

- There is no clear distinction between summary of results and their interpretation, and the transitions between topics (e.g. ROS, MMP, DFI) are weak and abruptly change focus?

  1. The discussion should emphasize what exactly is the new contribution of this research.

In Discusion was added new paragraph with contributions.

  1. In the methods section:

- Why did the study only include patients with ejaculate volumes above 1.6 mL, which limits the applicability of the results to men with low ejaculate volumes and does this not compromise the assessment of the effectiveness of MFSS in this group?

It is true that this inclusion criterion excluded some patients, but it was necessary because each sample had to undergo a standard sperm analysis, ROS detection and two separations: wim-up separations, and the MFSS method. MFSS, according to the protocol, requires the entry of 1 ml of ejaculate. When reducing the volume and adding media. The ratio would be violated and it would be difficult to compare the two methods if a different volume of ejaculate entered the system each time.

  1. Conclusion:

It is not stated what are the limitations of the MFSS method in patients with lower ejaculate volume or high levels of reactive oxygen species (ROS) and how this method can be optimized for these patient groups?

Limitationts of MFSS are now descibed in discussion.

The limitations of the current study need to be determined.

Limitation was added.

The English could be improved to more clearly express the research.

English was corrected

Round 2

Reviewer 2 Report (New Reviewer)

Comments and Suggestions for Authors

Your manuscript has undergone a very thorough revision, and its quality has improved significantly. I have only a few minor suggestions:

  • Line 358
    Instead of simply “4.1. Patients,” I recommend using “4.1. Study Design, Patient Eligibility Criteria, and Sample Division,” as this title better reflects the actual content of the section.

  • Lines 367–368
    The sentence starting with “Eligibility criteria were…” appears to overlap with the content of line 364. Please check whether this is redundant.

  • Lines 379 and 393
    You have used both “Sixty minutes after ejaculation” (line 379) and “60 min after ejaculation” (line 393). Please confirm whether the statement “Sixty minutes after ejaculation” was applied consistently to all samples—both those that liquified quickly and those that required a longer liquefaction period—and indicate if any adjustments were made for samples needing extra time.

  • Line 519
    Typically, the study’s limitations are presented in the final paragraph of the Discussion section rather than in the Conclusion. For instance, you might say, “This study has three limitations. First… Second…,” and so on. It would be more standard to follow that format.

  • “Author Response to Report”
    Except for clearly unnecessary details, please integrate your explanations from the “Author response to report” into the main text of the manuscript as much as possible. For example, the following statements are quite relevant and could be succinctly included:

    • “For verification of the homogenization process, we used triplicate aliquots for both concentration and motility analyses and performed several rounds of verification before sample evaluation.”

    • “In our case, the ejaculate is thoroughly homogenized first; we then measure its total volume, assess its pH, take a small sample for ROS assays, and obtain aliquots for WHO-based sperm evaluation.”

    • “For verifying our hypothesis, the patient’s identity was not crucial; rather, we focused on testing the effectiveness of the separation system across patients ranging from normal to infertile.”

    • “We used the older WHO 2010 reference because our lab has successfully relied on it for many years, and it remained a standard practice at the time we planned this study. Many centers still commonly use that edition today, and the differences from the 2021 version are not dramatic.”

Author Response

Your manuscript has undergone a very thorough revision, and its quality has improved significantly. I have only a few minor suggestions:

Comments1: Line 358

Instead of simply “4.1. Patients,” I recommend using “4.1. Study Design, Patient Eligibility Criteria, and Sample Division,” as this title better reflects the actual content of the section.

Response 1: Great idea thank you that's definitely better. It was changed

Comments2 Lines 367–368

The sentence starting with “Eligibility criteria were…” appears to overlap with the content of line 364. Please check whether this is redundant.

Response 2: That's right, it was changed

Comments3: Lines 379 and 393

You have used both “Sixty minutes after ejaculation” (line 379) and “60 min after ejaculation” (line 393). Please confirm whether the statement “Sixty minutes after ejaculation” was applied consistently to all samples—both those that liquified quickly and those that required a longer liquefaction period—and indicate if any adjustments were made for samples needing extra time.

Response 3: The 60-minute time was applied consistently to all patients. Fortunately, among the 48 patients analyzed, there was none with very viscous ejaculate, where sample liquefaction would have been impaired.

Comments4: Line 519

Typically, the study’s limitations are presented in the final paragraph of the Discussion section rather than in the Conclusion. For instance, you might say, “This study has three limitations. First… Second…,” and so on. It would be more standard to follow that format.

Response 4: We agree with this point. it was changed and moved to discussion

“Author Response to Report”

Except for clearly unnecessary details, please integrate your explanations from the “Author response to report” into the main text of the manuscript as much as possible. For example, the following statements are quite relevant and could be succinctly included:

Response 5: Thank you for pointing this out.

“For verification of the homogenization process, we used triplicate aliquots for both concentration and motility analyses and performed several rounds of verification before sample evaluation.”

“In our case, the ejaculate is thoroughly homogenized first; we then measure its total volume, assess its pH, take a small sample for ROS assays, and obtain aliquots for WHO-based sperm evaluation.”

Response 6: These two parts were added to the material and methods

“For verifying our hypothesis, the patient’s identity was not crucial; rather, we focused on testing the effectiveness of the separation system across patients ranging from normal to infertile.”

Response 6: This mention has been added to the discussion

“We used the older WHO 2010 reference because our lab has successfully relied on it for many years, and it remained a standard practice at the time we planned this study. Many centers still commonly use that edition today, and the differences from the 2021 version are not dramatic.”

Response 7: This was included into material and methods

Reviewer 3 Report (New Reviewer)

Comments and Suggestions for Authors

Accept in present form.

Author Response

Thank you very much for your help.

This manuscript is a resubmission of an earlier submission. The following is a list of the peer review reports and author responses from that submission.

Round 1

Reviewer 1 Report

Comments and Suggestions for Authors

This manuscript that entitled by “Comparison of swim-up and MFSS methods in selection of sperm for ICSI”, conducted a research study we assessed the effectiveness of Ca0 microfluidic chip compared to the swim-up (SU) technique with respect to oxygen radicals and spermiogram parameters. The findings stated that The Ca0 method is simple, frequently used in laboratories, and gives good results but does not provide much benefit over the swim-up. The manuscript is generally well-addressed and well-cited; however, I have some comments/suggestions.

Line 3: Please note that there are 2 abbreviations on the title (MFSS & ICSI) which keep it not clear and less comprehensible. I suggest rewriting the title with more informative words with understandable way.

Line 16: Please note that complete address information including zip code is required upon the journal guidelines which do not include all previous authors.

Line 17: I think the email address should be enough for the corresponding author.

Line 20: Before you start mentioning your materials and methods by " In this study, we assessed ", please mention why did you conducted this study and the importance of it then start the materials and methods used.

Line 33: I suggest rewriting the abstract to be more informative and comprehensible with a concise focused way as well. For example, the last sentence is missing something to get a complete conclusion for future work.

Line 34: Please verify the abbreviations of “IVF and ICSI”  as first time mentioned at the manuscript.

Line 51: The above information from the beginning of the introduction is well known for all as a basic information. Please rewrite to be more focused on the literature that related directly to the conducted study.

Line 89: The above paragraph at the end of introduction (from line 84-line 89) mentioned the main aim of the work but did NOT highlight the main conclusions. Please rewrite.

Line 90: I suggest moving the section of material and methods to be number 2 for easy tracking the study research as a whole story for readers. Just let us know the steps for what you did then show up the results.

Line 93: Please note that you mentioned at line 281 that: ejaculate volume greater than 1.6 ml were enrolled in the study.

Line 278: Please add the approval number and year to do this study by the ethics committee. Also, mention which country the University Hospital Brno located?

Line 280: Please mention the location of CERMED center and which country is located.

Line 293: Please rewrite the figure legend to be more informative and descriptive. Please verify the abbreviation of ROS. Also keep swim-up on the right side of each group for synchronization and easy tracking.

Line 299: Please add the catalog number for this Medium.

Line 311: Please add the catalog number for this Universal IVF medium.

Line 319: Please mention the name and the catalog number or producing company for these gravimetric measurements.

Line 327: please note that this sentence "The procedures followed WHO guidelines " repeated many times. Please use it one time as applied for the whole study.

Line 330: Please mention what stain did you used for staining the slides.

Line 352: how much is added to coat the slides with FITC-PSA. Please add reference for protocol that is used.

Line 360: Please move figure 5 with all his related data to the results section. That will be much better.

Line 397: Please rewrite the conclusion to include the aim of work and highlight your main findings by a simple and precise way.

References #17 and 34 are incomplete. Please revise.

Comments on the Quality of English Language

Major editing of English language required.

Author Response

Thank you very much for taking the time to review this manuscript. Please find the detailed responses below and the corresponding revisions/corrections highlighted/in track changes in the re-submitted files. All manuscript was carefully checked and language errors were corrected. All changes and corrections are visible in Word, tool all revisions.

Comments 1: Please note that there are 2 abbreviations on the title (MFSS & ICSI) which keep it not clear and less comprehensible. I suggest rewriting the title with more informative words with understandable way. Response 1: We agree with this comment. Tittle was changed.

Comments 2: Please note that complete address information including zip code is required upon the journal guidelines which do not include all previous authors Response 2: We agree with this comment. Addresses have been corrected.

Comments 3: Line 17: I think the email address should be enough for the corresponding author. Response 3: We agree. It was corrected.

Comments 4: Before you start mentioning your materials and methods by " In this study, we assessed ", please mention why did you conducted this study and the importance of it then start the materials and methods used. Response 4: We agree with this comment. Abstract was rewritten.

Comments 5: I suggest rewriting the abstract to be more informative and comprehensible with a concise focused way as well. For example, the last sentence is missing something to get a complete conclusion for future work. Response 5: Thank you for pointing this out. Abstract was completely rewritten.

Comments 6: Please verify the abbreviations of “IVF and ICSI”  as first time mentioned at the manuscript Response 6: We agree it was changed.

Comments 7: The above information from the beginning of the introduction is well known for all as a basic information. Please rewrite to be more focused on the literature that related directly to the conducted study. Response 7: We agree with this comment. Introduction was changed.

Comments 8: The above paragraph at the end of introduction (from line 84-line 89) mentioned the main aim of the work but did NOT highlight the main conclusions. Please rewrite. Response 8: We agree it was rewritten.

Comments 9: I suggest moving the section of material and methods to be number 2 for easy tracking the study research as a whole story for readers. Just let us know the steps for what you did then show up the results. Response 9: Thank you for pointing this out, but this is not possible. After my asking to editor he recommend me keep this format which follows to Journal's Regulations.

Comments 10: Line 93 Please note that you mentioned at line 281 that: ejaculate volume greater than 1.6 ml were enrolled in the study. Response 10: Yes mean volume was 7.19 but minimum for including to the study was 1.6 mL.

Comments 11: Please add the approval number and year to do this study by the ethics committee. Also, mention which country the University Hospital Brno located? Response 11: Thank you for pointing this out, now is it correct.

Comments 12: Please mention the location of CERMED center and which country is located Response 12: Corrected in chapter 2.1.Patients

Comments 13: Please rewrite the figure legend to be more informative and descriptive. Please verify the abbreviation of ROS. Also keep swim-up on the right side of each group for synchronization and easy tracking. Response 13: We agree, Figure was changed, legend was rewritten.

Comments 14: Please add the catalog number for this Medium. Response 14: Added

Comments 15: Please add the catalog number for this Universal IVF medium. Response 15: Added

Comments 16: Please mention the name and the catalog number or producing company for these gravimetric measurements Response 16: Corrected

Comments 17: please note that this sentence "The procedures followed WHO guidelines " repeated many times. Please use it one time as applied for the whole study. Response 17: We agre now is it corrected and WHO is mentioned only in chapter 4.1.

Comments 18: Please mention what stain did you used for staining the slides. Response 18: thank you, it was Diff-Quick staining now is it in chapter 4.3

Comments 19: how much is added to coat the slides with FITC-PSA. Please add reference for protocol that is used. Response 19: Thank you for pointing this out, now it was added informations including reference.

Comments 20: Please move figure 5 with all his related data to the results section. That will be much better. Response 20: We agree, figure was moved to results.

Comments 21: Please rewrite the conclusion to include the aim of work and highlight your main findings by a simple and precise way. Response 21: Conlusion was completely rewritten.

Comments 22: 17 and 34 are incomplete. Please revise. Response 22: Thank you for pointing this out, now is it correct.

Corrections are marked red and all changes are visible in Word sll revisions tool

Reviewer 2 Report

Comments and Suggestions for Authors

Although authors compared some parameters about swim-up and microfluidic sperm selection systems (MFSS) methods in selection of sperm for intracytoplasmic sperm injection, the design and methods are so similar with reference 14 and sample numbers are small. There are few new ideas and perspectives.

The parameter data about normozoospermic (n=25) and non-normozoospermic (n=20) should be added in Table 1 and 2.

MFSS and CA0 are mixed in sentences and figures, should be rechecked and unified as MFSS.

The staining data of sperm morphology should be added to Figure 1.

Figure 4 is nonsense and should be removed.

The English language and style need editing. The abbreviated words should be checked again.

Comments on the Quality of English Language

The English language and style need editing. The abbreviated words should be checked again.

Author Response

Thank you very much for taking the time to review this manuscript. In this study, unlike the cited ref 14, we carefully monitor the level of ROS in the seminal plasma in order to prevent the risk of secondary damage to sperm during the separation process. At the same time, unlike the aforementioned study, we evaluate slightly different parameters and monitor, for example, the level of mitochondrial membrane potential. Please find the detailed responses below and the corresponding revisions/corrections highlighted/in track changes in the re-submitted files.

Comments 1: The parameter data about normozoospermic (n=25) and non-normozoospermic (n=20) should be added in Table 1 and 2. Response 1: We agree with this comment. Table 1 was changed. In the case of specific parameter values, this is a difficult comparison because in the group of non-normozoospermic patients there is big heterogeneity between patients in sperm parameters.

Comments 2: MFSS and CA0 are mixed in sentences and figures, should be rechecked and unified as MFSS. Response 2: Thank you for this point, now is it corrected as MFSS.

Comments 3: The staining data of sperm morphology should be added to Figure 1.

Response 3: Thank you for this point, morphology data are in Figure 1 visualized as Normal sperm morphology.

Comments 4: Figure 4 nonsense and should be removed. Response 4: Figure 4 was removed from manuscript.

Comments 5: The English language and style need editing. The abbreviated words should be checked again. Response 5: We agree, all manuscript was carefully checked and language errors were corrected. All changes and corrections are visible in Word, tool all revisions

Round 2

Reviewer 1 Report

Comments and Suggestions for Authors

The manuscript is improved. Thank You!.

Comments on the Quality of English Language

Minor English language editing required.

Author Response

Thank you very much for taking the time to review this manuscript. Please find the detailed responses below and the corresponding revisions/corrections highlighted/in track changes in the re-submitted files..

Comments 1: The manuscript is improved. Thank You! Response 1: We agree with your comments, now is it better.

Comments 2: Minor English language editing required. Response 2: All manuscript was carefully checked and language errors were corrected. All changes and corrections are visible in Word, tool all revisions

Corrections are marked red and all changes are visible in Word revisions tool

Reviewer 2 Report

Comments and Suggestions for Authors

The parameter data about normozoospermic (n=25) and non-normozoospermic (n=20) should be added in Table 1 and 2, but not only the number of patients.

MFSS and CA0 are still mixed in Figures.

The staining data of sperm morphology from normozoospermic and non-normozoospermic should be added to Figure 1.

There are few new ideas and perspectives, and sample numbers are small.

Comments on the Quality of English Language

The English could be improved to more clearly express the research.

Author Response

Comments 1: The parameter data about normozoospermic (n=25) and non-normozoospermic (n=20) should be added in Table 1 and 2, but not only the number of patients. Response 1: Table 1 was changed and all parameters about normozoospermic and non-normozoospermic patients was add in table 3 and table 4 as a neat ejaculate before separation.  

Comments 2: MFSS and CA0 are still mixed in Figures. Response 2: Thank you, now is it corrected throughout the manuscript including Figures.

Comments 3: The staining data of sperm morphology from normozoospermic and non-normozoospermic should be added to Figure 1.

Response 3: Thank you for this point, morphology data are in Figure 1 newly presented in all, normozoospermic and non-normozoospermic patients.

Comments 4: There are few new ideas and perspectives, and sample numbers are small. Response 4: In this manuscipt we compared MFSS and swim-up technic and we check ROS concentrationts. In this study, we monitor the level of ROS in the seminal plasma in order to prevent the risk of secondary damage to sperm during the separation process. At the same time, we evaluate parameters like acrosomal status mitochondrial potential and DNA fragmentation. Our data indicate differences in seprartion mainly in non-normozoospermic patients. Is it quite new approach and original results in human sperm. According your recommendation regrading small sample - during 2 weeks we intensively prepared 10 new samples (5 normozoospermic and 5 non-normozoospermic) which are now newly included in the manuscript.